# Locally coupled electromechanical interfaces based on cytoadhesion-inspired hybrids to identify muscular excitation-contraction signatures

Pingqiang Cai [1], Changjin Wan [1], Liang Pan [1], Naoji Matsuhisa [1], Ke He [1], Zequn Cui [1], Wei Zhang [1], Chengcheng Li[1], Jianwu Wang [1], Jing Yu [1], Ming Wang[1], Ying Jiang [1], Geng Chen [1] & Xiaodong Chen [1✉]

Coupling myoelectric and mechanical signals during voluntary muscle contraction is paramount in human–machine interactions. Spatiotemporal differences in the two signals intrinsically arise from the muscular excitation–contraction process; however, current methods fail to deliver local electromechanical coupling of the process. Here we present the locally coupled electromechanical interface based on a quadra-layered ionotronic hybrid (named as CoupOn) that mimics the transmembrane cytoadhesion architecture. CoupOn simultaneously monitors mechanical strains with a gauge factor of ~34 and surface electromyogram with a signal-to-noise ratio of 32.2 dB. The resolved excitation–contraction signatures of forearm flexor muscles can recognize flexions of different fingers, hand grips of varying strength, and nervous and metabolic muscle fatigue. The orthogonal correlation of hand grip strength with speed is further exploited to manipulate robotic hands for recapitulating corresponding gesture dynamics. It can be envisioned that such locally coupled electromechanical interfaces would endow cyber–human interactions with unprecedented robustness and dexterity.

---

[1] Innovative Centre for Flexible Devices (iFLEX), Max Planck–NTU Joint Lab for Artificial Senses, School of Materials Science and Engineering, Nanyang Technological University, 50 Nanyang Avenue, Singapore 639798, Singapore. ✉email: chenxd@ntu.edu.sg

During voluntary muscle contraction, myoelectric stimuli (the neurologically activated action potential) are transduced into mechanical responses (the twitching of myofibers), known as the excitation–contraction coupling[1]. Versatile methods have been presented to capture neuromuscular performance by retrieving the myoelectric and mechanical signals. Recent advances in novel surface electromyogram (sEMG) electrodes[2–4] and skin-mountable strain sensors[5–9] have therefore been implemented in a variety of applications, such as the manipulation of prosthetic limbs[10,11], human–machine interactions[12,13], health monitoring[14–16], and the prognosis of neuromuscular disorders[17,18]. Critically, the fidelity and predictive power of independent sEMG or strain sensors for evaluating muscular activities can be compromised by the spatiotemporal differences[19,20] in the pattern of myoelectric triggers and such mechanical responses as the myofiber shortening. In line with this, the clinical utility of sEMG alone is less capable of distinguishing between neuropathic or myopathic conditions for the diagnosis of neuromuscular disorders[21].

Hence, local identification of the muscular excitation–contraction signatures is on demand, comprising of coupled myoelectric signals (e.g., sEMG) and mechanical responses (e.g., superficial skin strain). Hydrogel ionotronic devices are promising candidates due to the intrinsic similarities of ionic hydrogels with the soft and wet living tissues that transmit electrophysiological signals through mobile ions[22,23]. They can couple electrons in metal conductors and ions in hydrogels to deliver multifunctional human–machine interfaces[24–26]. A few strategies have therefore been presented to achieve strong interfacial adhesion between ionic gels and microstructured metallic films[27,28] and metal wires[29]. However, the challenge for hydrogel ionotronic devices with robust strain sensitivity remains unresolved, frustrating the identification of muscular excitation–contraction coupling[24,27,29]. Ideal locally coupled electromechanical interfaces should be characterized by low interfacial impedance, high strain sensitivity, and tough interfacial bonding[30–32]. To address these issues, we turn to a strategy that is adopted by adherent mammalian cells. Adherent cells (e.g., fibroblast and epithelial cells) adhere to the extracellular matrix via discrete focal adhesions[33], namely the microscale transmembrane machinery that mechanically links the intracellular polymeric microfilaments to the extracellular matrix. Such mechanical links remain robust under cyclic stretch, as their cohesion strength is enhanced with specific ligand binding[34].

Inspired by such mechanically robust cytoadhesion, we develop a quadra-layered ionotronic hybrid integrating the ionic hydrogel and strain-sensitive double metallic nanofilm onto the elastomer with "adhesion plaques" and tough bonding at the interface. The synergy of interfacial tough bonding and contact splitting herein endow the hybrid (named as *CoupOn*) with strong interlayer adhesion (~400 N m$^{-1}$). Given the interface with the electronic/ionic coupling that is highly sensitive to both electrophysiological and mechanical signals, *CoupOn* is capable of identifying the excitation–contraction signatures of forearm muscles by locally coupling the sEMG signal and skin strain. The extracted signatures can be well correlated with the dynamics (i.e., amplitude, strength, and speed) of hand grip gestures and finger flexions. These merits make such locally coupled electromechanical interfaces promising for next-generation multifunctional cyber–human interfaces.

## Results

### Mechanically integrated hybrid interfaces of *CoupOn*. To counteract the spatiotemporal differences in action potentials and the triggered myofiber shortening during voluntary muscle

contraction (Fig. 1a and Supplementary Note 1), we developed a quadra-layered ionotronic hybrid capable of strain sensing and sEMG recording (Fig. 1b). Resistivity-based stretchable strain sensors can be developed by depositing a conductive nanofilm on elastomeric substrates[35,36]. The electrical resistance of thin films drastically increases with the applied strain that leads to the propagation of pre-existing microcracks[37,38]. We propose that these microcracks might also allow the penetration of pre-gel solution, thereby the formation of cytoadhesion-like microstructures (Fig. 1c, d). Meanwhile, we propose that a double metallic nanofilm, comprising of a layer of brittle metal nanofilm (i.e., titanium) and a ductile metal nanofilm (i.e., gold), could achieve both high sensitivity to mechanical strains and high stretchability. First, gold nanofilm (thickness of ~40 nm) was thermally deposited onto the poly-dimethylsiloxane (PDMS) film (thickness of ~40 μm; crosslinker to monomer ratio 1 : 10) at the rate of 10 Å/s. Interestingly, "holes" with smashed Au speckles were observed on the Au nanofilm (Supplementary Fig. 1a), in addition to commonly reported tri-branched microcracks[37]. Then, a thin layer of titanium nanofilm (thickness of ~10 nm) was sputtered onto the obtained microcracked Au nanofilm. Energy-dispersive X-ray spectroscopy showed that Ti could reach the elastomer through the microcracks and "holes" of the Au nanofilm (Supplementary Fig. 1b). By varying the elastomer stiffness and adhesiveness, we found that the condition for forming such "holes" with smashed Au speckles seemed stringent. Alternatively, the "hole" size can be modulated by masking the elastomer with discrete water-soluble polyvinyl alcohol (PVA) disks before Au deposition, which were dissolved afterward (Supplementary Fig. 2).

Subsequently, the linker 3-(trimethoxysilyl) propyl methacrylate (TMSPMA)[39] was introduced to form tough bonding between the ionic gel and the double metallic nanofilm (Supplementary Fig. 3). Raman and FTIR spectra confirmed the salinization of TMSPMA onto Ti surface and (Supplementary Figs. 4 and 5). Tough hydrogel was synthesized according to a previous report[40], but using carbonate calcium nanopowders as the physical crosslinkers. This allowed a slowed gelation of the alginate network (Supplementary Fig. 6), thereby the prolonged penetration of the pre-gel solution under vacuum. The obtained tough gel showed a high stretchability bearing up to 2130% strain and the Young's modulus of 31 kPa (Supplementary Fig. 7) similar to soft biological tissues[41,42]. The successful anchorage of tough gel onto the silanized Ti surface was also suggested by Fourier-transform infrared (FTIR) spectra (Supplementary Fig. 5). To further demonstrate the tough bonding between the gel and double metallic nanofilm, the tough gel was sandwiched on double metallic nanofilm deposited on a glass slide, following the aforementioned bonding procedures. With the covalent bonding of the polyacrylamide network and surface-bound TMSPMA, the tough gel could readily peel double metallic nanofilm off the glass (Supplementary Fig. 8a). The obtained metal-gel film could bear harsh squeeze and press even when submerged in water (Supplementary Fig. 8b).

Next, the adhesion of the elastomer layer with double metallic nanofilm and tough gel was investigated. The elastomer of the *CoupOn* hybrid was removed after infiltrating the ionic gel with epoxy resins of graded concentration[43]. This allowed visualization of the gel penetration through the microcracked metallic nanofilm. Consistent with our hypothesis, nanoscale "adhesion plaques" out of the metallic nanofilm were observed, which evenly distributed at the interface (Fig. 2a and Supplementary Fig. 9). The adhesion plaques had an average projected area of 0.4 μm$^2$ and an average minimum distance of 6.8 μm (Fig. 2b, c). In addition, atomic force microscopy images confirmed that these adhesion plaques were higher than the peripheral microcracked

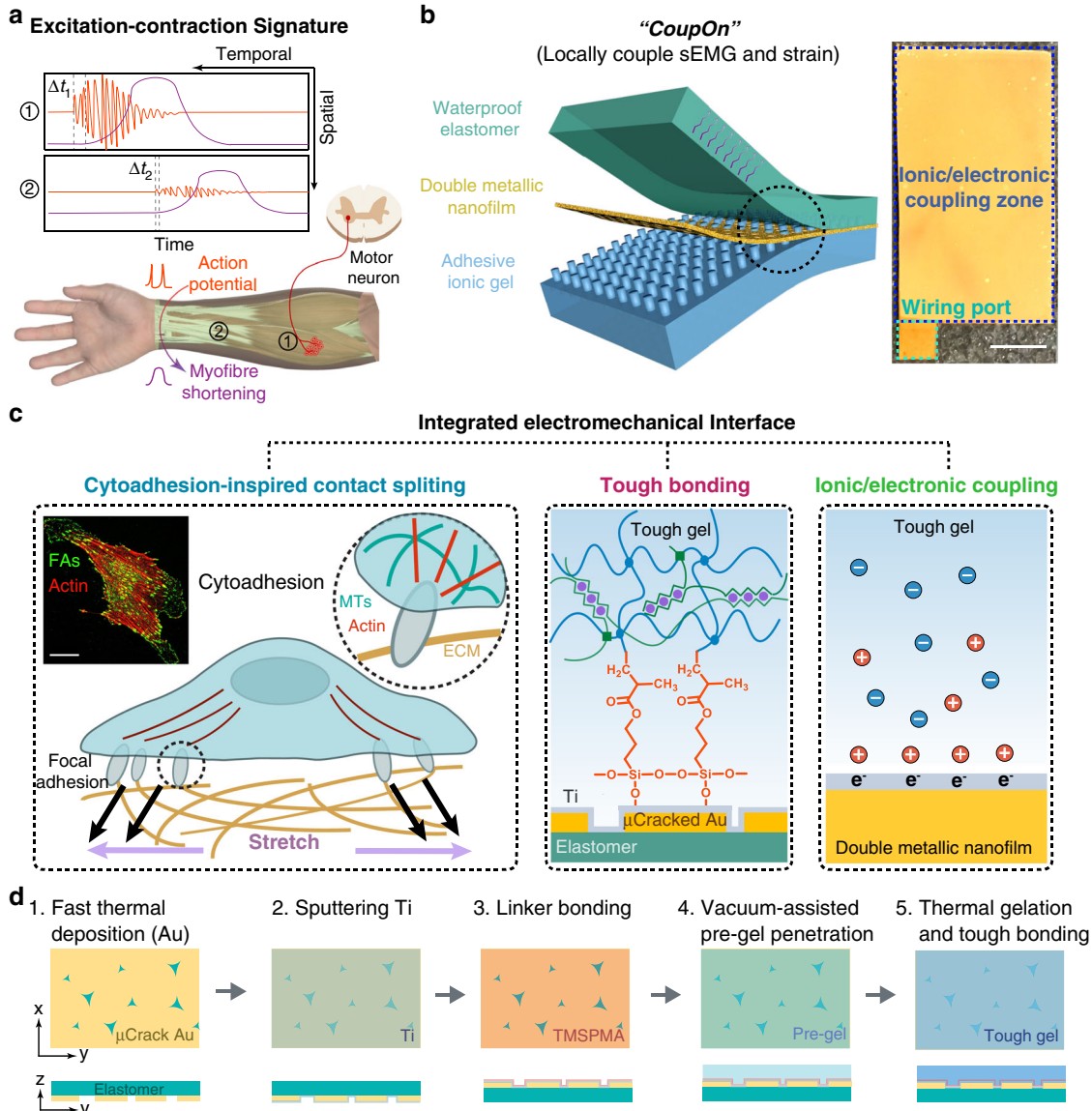

**Fig. 1 Cytoadhesion-inspired quadra-layered hybrids. a** Schematic shows the temporal and spatial differences in the patterns of myoelectric excitation and mechanical contraction in the excitation–contraction coupling process during voluntary muscle contraction. **b** Schematic of the integrated electromechanical interface (i.e., *CoupOn*). Right photograph shows one fabricated *CoupOn* hybrid with the ionotronically conductive zone and the wiring port annotated. The adhesive ionic hydrogel retrieves sEMG signals and the stretchable double metallic nanofilm functions as the resistive strain sensor while transmitting the electrical signals for readouts. The waterproof elastomer mitigates water loss from the ionic hydrogel. Scale bar, 5 mm. **c** Schematic of strong interlayer adhesion at the integrated electromechancial interface that combines the cytoadhesion-inspired contact splitting and tough interfacial bonding, illustrating dash circle in **c**. Inset, confocal fluorescent images showing an adherent fibroblast labeled with actin microfilaments (red) and focal adhesions (green). Scale bar, 30 μm. **d** Schematic of the fabrication process of the hybrid *CoupOn* in five steps.

metallic nanofilm (Fig. 2d). It appeared to be the "holes" that dominated the penetration of pre-gel solution and therefore the formation of adhesion plaques. By contrast, microcracks allowed limited penetration of the pre-gel, as evident from the height profile and phase mapping (Fig. 2e). The adhesion plaques had an average out-of-plane height of about 45 nm (relative to metallic nanofilms; Fig. 2f, g), which might allow intimate and split contact of ionic gel with the elastomer. Such adhesion plaques could not only allow direct bonding of ionic gel with the elastomer via functionalized Ti surface but might also dissipate energy when subjected to strain due to the contact splitting mechanism[44,45]. Such cytoadhesion-inspired hybrid interface showed a strong interlayer adhesion at the magnitude around 400 N m⁻¹, which resulted in the gel fibrils formation during the peel-off test (Fig. 2h). The strong interlayer adhesion also enabled the formation of consistently thick tough gel without delamination when submerged in water (Supplementary Fig. 10a). In addition, the flat nature of the metallic nanofilm and the tough gel was well kept by their intimate bonding at the interface, upon the removal of the elastomer layer after freeze-drying (Supplementary Fig. 10b). By contrast, the absence of either specific tough bonding or discrete adhesion plaques would significantly compromise the interlayer adhesion (Fig. 2i). In short, the strong interlayer adhesion was achieved by combining tough interfacial bonding between the double metallic nanofilm and the tough gel, and the contact splitting mechanism. This combination shares two features of cytoadhesion, namely the specific recognition and transmembrane structures of cellular focal adhesions[46].

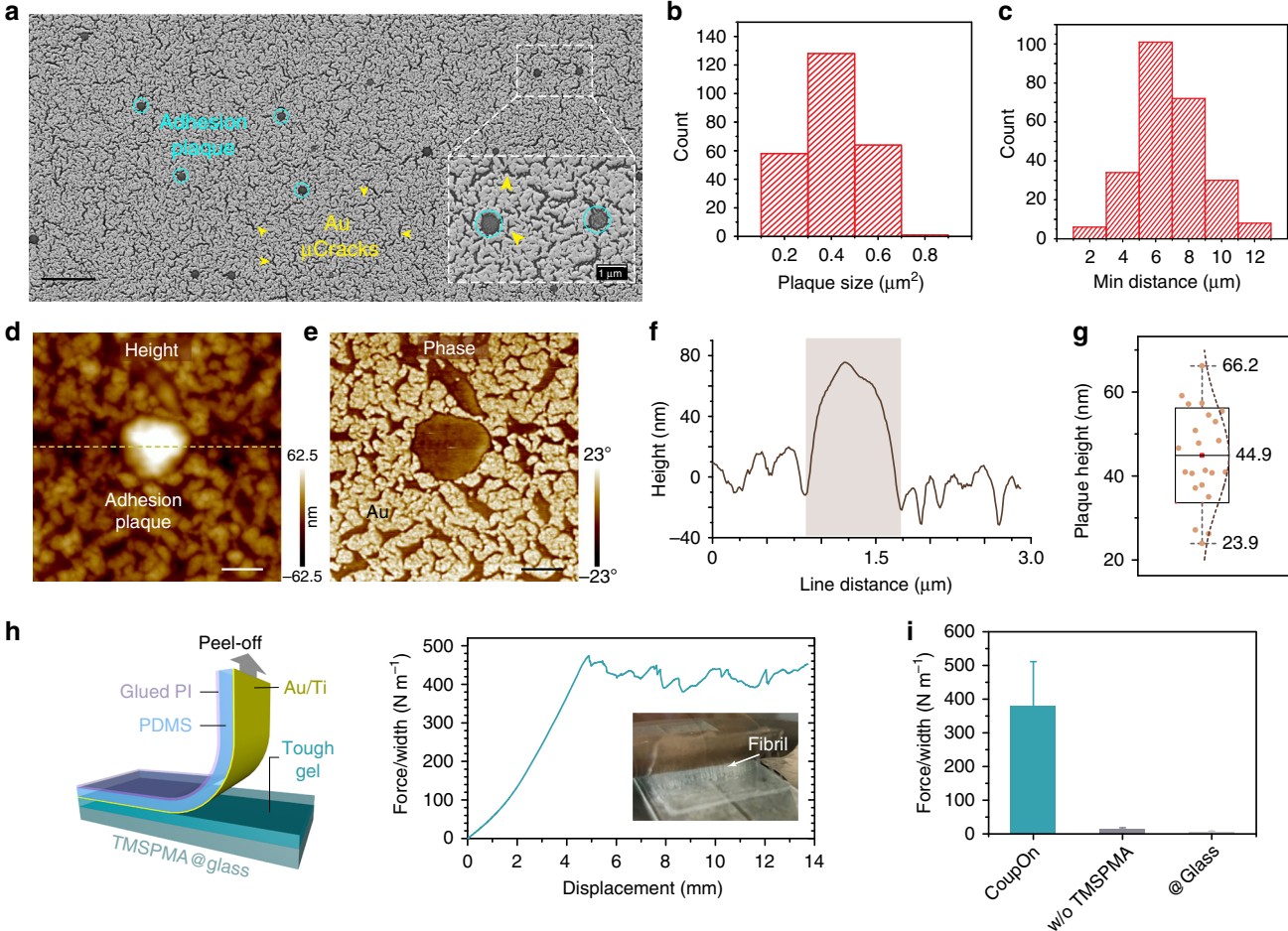

**Fig. 2 Formation of cytoadhesion-like plaques endowed the strong interlayer bonding. a** SEM image shows the formation of discrete hydrogel-elastomer adhesion plaques (cyan circles), which are out of the metallic nanofilm through the microcracks (yellow arrowheads) as shown in the inset (dash white rectangle). Representative of three samples after elastomer removal. Scale bar, 5 μm. **b** Histogram plot shows the size distribution of adhesion plaques. **c** Histogram plot shows the distribution of minimum distances between adhesion plaques in proximity. Two hundred and forty-six adhesion plaques are analyzed. **d** AFM height profile shows the out-of-plane adhesion plaques (white zone). **e** AFM phase image showed a clear distribution of metallic nanofilm and microcracks filled by the tough gel. Representative image of three samples. Scale bar, 0.5 μm. **f** Line plot shows the height profile along the dash line in **d**. **g** Box chart shows the height of adhesion plaques compared to the peripheral metallic nanofilm. Twenty-two adhesion plaques are analyzed. Upper and lower whisker denote the max and min data value, upper and lower quartile denote ± SD, middle quartile denotes the mean. **h** Schematic illustrates the 90° peel-off test setup and line plot shows the interlayer adhesion between the metallic nanofilm/elastomer and tough gel. Inset: Photograph shows the formation of fibrils during the peel-off test. **i** Bar chart shows the adhesion strength of different interfaces. "w/o TMPSPMA" group refers to non-silanized metallic nanofilms on PDMS, "@Glass" group refers to Au/Ti nanofilms on the glass that forms non-microcracked Au/Ti nanofilms. Representative of three to five samples with a dimension of 1.5 cm by 4.0 cm. Data are presented as mean ± SD.

**Retrieval of myoelectric and strain signals**. Before leaping into the electromechanical coupling, the mechanical and electrical performance of *CoupOn* as a patchable electrode was investigated independently. *CoupOn* showed adhesive strength comparable to commercial adhesive sEMG electrodes (i.e., *Vitrode* F150ML, Nihon Kohden). The standalone adhesive gel could be stretched to six times of its original length, while remaining adherent to the skin (Supplementary Fig. 11a), probably due to interfacial fluid transport[47]. *CoupOn* also showed repeatable adhesion to the porcine skin without compromising the adhesion strength (around ~15 N m$^{-1}$) within five peel-off tests (Supplementary Fig. 11b). Such capability of repeatable adhesion would allow the repositioning of the *CoupOn* in practical applications. It is noteworthy that the adhesion strength is much higher than the driving forces (~8.4 N m$^{-1}$) needed for delaminating the *CoupOn* from the skin under 50% strain (Supplementary Note 2).

Interestingly, the adhesiveness and softness of the ionic gel also significantly promoted the conformal contact of *CoupOn* on skins, allowing the clear visualization of such skin microstructures as fine wrinkles (Fig. 3a). Such conformal contact would contribute the minimizing both background noise and motion artifacts. By contrast, the thin elastomer with double metallic nanofilm delivered poor contact with the skin, though the thickness is much lower (~40 μm vs. ~140 μm of *CoupOn*). It suggests that interfacial softness and adhesiveness might be dominant over film thickness[12] for achieving conformal contact. In addition, the integration with waterproof elastomer could effectively mitigate dehydration of the ionic gel, which reduced the water loss from 90% by weight to 15% after 6 h of wearing and exposure to the ambient environment (Supplementary Fig. 12).

The sensitivity of a strain sensor is defined by the gauge factor (GF; Eq. (1)) referring to the relative resistivity change with

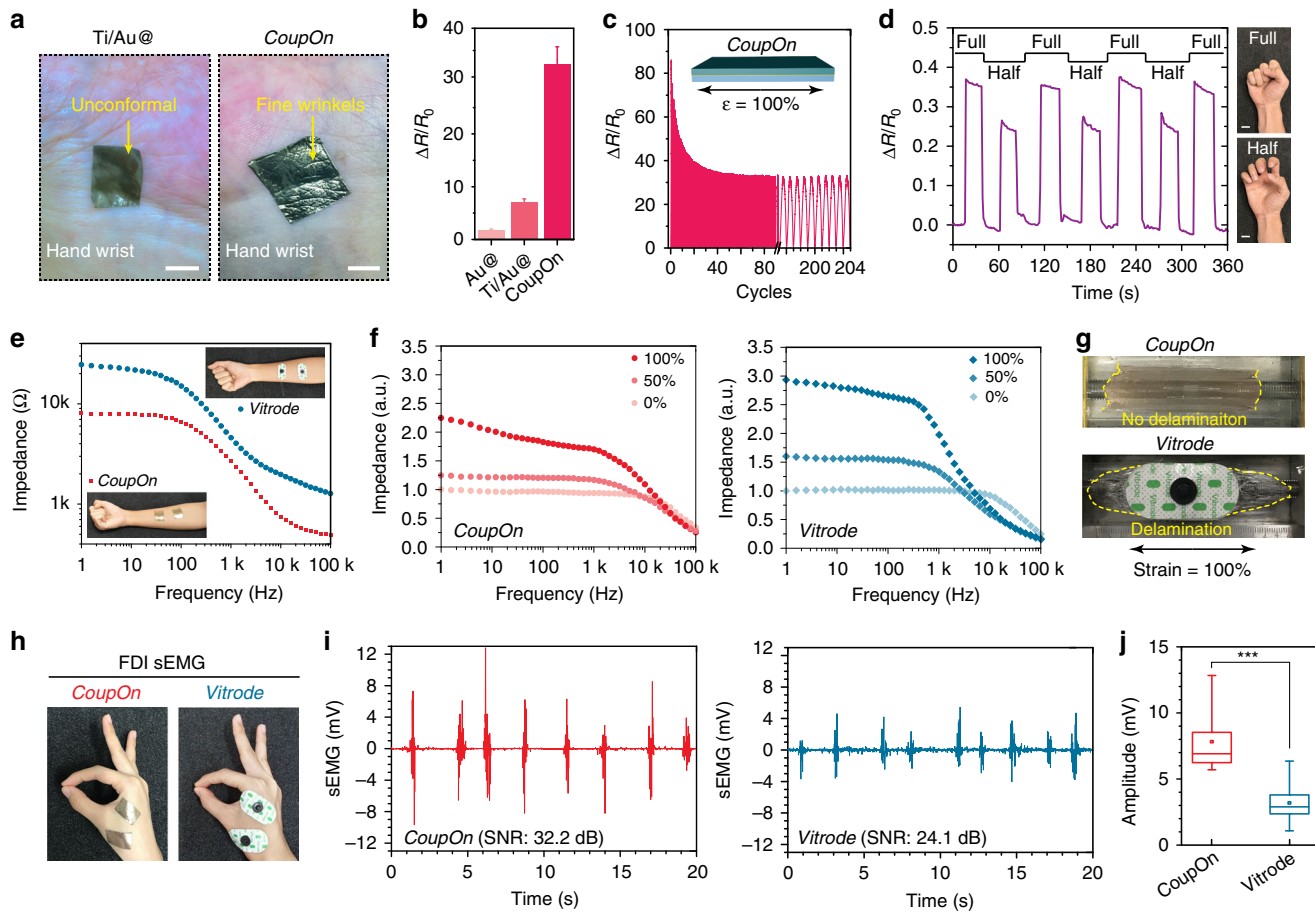

**Fig. 3 Mechanical and electrical characterizations of CoupOn. a** Photographs show the conformal contact of CoupOn with the skin of hand wrist enabling the observation of underlying fine wrinkles observed, in contrast to Ti/Au@elastomer in loose and unconformal contact. Scale bar: 1 cm. Representative of three tests. **b** Bar chart shows that both the introduction of 10 nm titanium layer and 100 μm hydrogel increase the gauge factor when bearing 100% strain. **c** Plot shows the resistivity change *CoupOn* in over 200 cycles bearing 100% strain. **d** Plot shows that resistive strain sensing on the forearm muscle can discern full and half fist clenching. Scale bar, 2 cm. **e** Dot plot shows the low skin contact impedance of *CoupOn* compared to its commercial counterpart *Vitrode*. **f** Dot plot shows the change of contact impedance of *CoupOn* and *Vitrode* when subjected to 0%, 50%, and 100% strain respectively, using ionic PVA films as the skin substitute. **g** Photographs show that *CoupOn* has no interlayer delamination when stretched, in contrast to commercial *Vitrode* with obvious delamination of the gel from the supporting fabric. Both hybrids adhere to the PVA film. **h** Photographs show the setup for detecting first dorsal interosseous (FDI) sEMG. **i** Plots show that the retrieved FDI sEMG using the *CoupOn* and commercial *Vitrode*. *CoupOn* offers a much higher SNR (32.2 dB) than commercial *Vitrode* (24.1 dB). **j** Box plot shows that the amplitude of signals retrieved by *CoupOn* is higher than *Vitrode*. Representative of four samples. Data are presented as mean ± SD. ***$P \leq 0.001$ (Student's $t$-test).

respect to the bearing strain $\varepsilon$:

$$GF = \frac{\Delta R}{\varepsilon R_0} \qquad (1)$$

It turned out the sequential introduction of Ti and tough gel increased the GF of pure microcracked Au nanofilm, from 2.4 to 7.9 and 34.2, respectively (Fig. 3b and Supplementary Fig. 13a). Despite the increased GF of *CoupOn*, the stretchability was not compromised, which was equally capable of bearing 100% strain for over 200 cycles (Fig. 3c; Supplementary Fig. 13b-c). Given the established strain sensitivity, the capability of *CoupOn* to monitor the skin strain caused by the contracting forearm muscle during hand grips was investigated. *CoupOn* was placed on the belly of flexor digitorum superficialis (FDS). It was able to distinguish half and full fist closure with the relative resistivity change ($\Delta R/R_0$) reaching ~0.25 and ~0.35, respectively, even with no tightening force applied in both cases (Fig. 3d).

Low skin contact impedance is crucial for retrieving high-quality sEMG signals. *CoupOn* showed a much lower impedance

with skin (the whole impedance comprises of the contact impedance and skin's impedance) at full frequency range and a low impedance of 8 kΩ at 1 Hz frequency, compared with 25 kΩ of the commercial *Vitrode* (contact area ~8 cm², the center-to-center distance ~5 cm; Fig. 3e). To investigate whether such contact impedance would be influenced dramatically when subjected to strain, a pair of *CoupOn* and *Vitrode* (F150ML), respectively, was attached to the skin substitute (PVA film containing 2 wt.% $CaCl_2$). The underlying PVA film was then stretched to 150% and 200% of the original length. *CoupOn* even underwent a lower increase in the impedance, in addition to the lower initial impedance (Fig. 3f). Under the 50% and 100% strain, the impedance of *CoupOn* at 1 Hz increased to 1.2 folds and 2.2 folds, respectively, whereas that of *Vitrode* increased to 1.7-folds and 2.9-folds. The relatively lower change in the impedance could allow relatively stable retrieval of electrophysiology signals when bearing mechanical strain. It is noteworthy that the gel delaminated from the fabric of *Vitrode*, but not shown in the stretched *CoupOn* (Fig. 3g), which suggested the strong interlayer adhesion.

To further evaluate *CoupOn* for retrieving electrophysiology signals, sEMG of first dorsal interosseous (FDI) muscles was recorded (Fig. 3h). FDI sEMG is one clinically adopted indicator for the diagnosis of Parkinson's disease. FDI muscle is driven by small motor units; hence the myoelectric triggers are low in such movements as thumb-index finger pinching. FDI sEMG retrieved by *CoupOn* showed a much higher signal-to-noise ratio than that by *Vitrode* (Fig. 3i), which was 32.2 dB and 24.1 dB, respectively, according to a recent algorithm[48]. Likewise, the average peak amplitude of signals retrieved by *CoupOn* was 6.8 mV, in contrast to 2.6 mV of those retrieved by *Vitrode* (Fig. 3j). It suggests that *CoupOn* is capable of delivering sEMG signals of high quality, superior to commercial counterparts.

**Identifying local muscular excitation and contraction**. Coupling the mechanical effectiveness and underlying neural drive is of significant importance in the improved understanding of excitation–contraction signatures of dynamic muscle activities. FDS is the intermediate muscle in the forearm, which flexes proximal interphalangeal and metacarpophalangeal joints of the index, middle, ring and little fingers. It also forms four tendons passing through the carpal tunnel of the wrist into the four fingers[49]. Hence, identifying the local muscular excitation–contraction signatures of FDS (Fig. 4a) would be promising in recognizing hand gestures. Above all, it was confirmed that resistivity change in *CoupOn* caused little influence on sEMG retrieval. In addition to the fact that the range of resistivity in metallic nanofilms was ~2 orders lower than the range of its contact impedance with the skin, it was further revealed that passive skin deformation by external touch at the proximity of *CoupOn* was simply characterized with sharp peaks in the mechanical strain, but no obvious sEMG spikes (Fig. 4b).

To validate the capability of *CoupOn* in identifying muscular excitation–contraction signatures of FDS muscles, 11 subjects were recruited with varying maximum grip forces, forearm girth, as well as different genders and ages (Fig. 4c and Supplementary Table 1). By combining the muscle belly strain (hybrid 1 in Fig. 4a) and sEMG signals from FDS muscle (Fig. 4d), *CoupOn* could distinguish a fist closure with minimal voluntary contraction (defined as minFist) from a fist clenching with maximal voluntary contraction (defined as maxFist). In a minFist, subject 9 showed little sEMG signals, though the relative resistivity ($\Delta R/R_0$) changed by ~0.25. In a maxFist, strong sEMG signals were observed with the relative resistivity changed by ~0.27. It is noteworthy that subject 5 showed sEMG signals at a low amplitude during a minFist with a grip dynamometer (0 kg grip in Fig. 4g). In fact, the closure of a minFist also involved the flexion of distal interphalangeal joints of fingers that were driven by flexor digitorum profundus (FDP) in the deep layer[49]. Meanwhile, motor unit type I would be firstly recruited at lower force levels (e.g., minFist), generating relatively low sEMG. The amount of motor unit type I in both FDS and FDP muscles can vary among subjects, causing the inter-subject difference in sEMG signals during a minFist. Therefore, identifying the alternating minFist and maxFist could be challenged by either single signal of the mechanical strain or the myoelectric trigger for those with sensitive motor unit type I or deeper FDS and FDP muscles. Interestingly, higher spikes were found at the initial stage of the tightening phase, followed by the much lower spikes. This was suggestive of the nervous muscle fatigue[50], although the gesture of fist tightening remained.

In addition, the coupling of mechanical events (e.g., muscle contraction and force generation) and the myoelectric excitation can vary with regards to types of muscles, magnitude, and orientation of muscle contraction. Hence, such electromechanical coupling is equally sensitive to the region of interest. When adopting the *CoupOn* (hybrid 2) on FDS tendons as the strain sensor, it showed relatively complicated excitation–contraction signatures during a maxFist (Fig. 4e). Taking subject 9 as an example, the retrieved excitation–contraction signatures could be dissected into four phases, corresponding to the following: (I) fist closing, (II) fist tightening, (III) fist relaxing, and (IV) fist opening. Interestingly, a small peak followed the drastic drop in the mechanical strain during the phase of fist opening, in contrast to little characteristics in sEMG signals. This was probably due to the complex superficial deformation on the skin above FDS tendons. More interestingly, the complexity of strain curves varied remarkably among different subjects (Supplementary Fig. 14), which probably aroused from the inter-subject variability in tendon depth and forearm muscle anatomy. It also implied that the coupling of tendon strain sensing and sEMG of FDS muscle can be promising for personal precision healthcare monitoring and rehabilitation.

Subsequently, the quantitative aspect of the electromechanical coupling during voluntary contraction was further investigated. The excitation–contraction signatures were extracted from the 11 subjects conducting standardized tasks using a grip dynamometer at different force levels, i.e., 0, 10, 20, and 30 kg (Fig. 4f). Although an increase with the force level was observed in both sEMG and muscle belly strain, the trend of the increase appeared divergent (Fig. 4g and Supplementary Fig. 15). It seemed that sEMG recording was more sensitive to hand grips at higher force levels (e.g., 20 and 30 kg), being consistent with previous reports[51,52]. On the contrary, strain sensing could be quite responsive to hand grips at lower force levels (e.g., 0 and 10 kg), while the increase of strain from 20 kg grips to 30 kg grips seemed relatively moderate (Fig. 4h). Such divergence in the change of sEMG and strain appeared consistent over the 11 subjects (Fig. 4i). Their coupling efficiency seemed to decrease with grip forces, as evident from the ratio of normalized strain to sEMG signals (Fig. 4j). The degree of such divergence and coupling efficiency decrease also differed between subjects 1–3 and subjects 4–11 (Supplementary Figs. 16 and 17), probably due to the different forearm girth and maximal grip forces (Supplementary Fig. 18). Given such divergence, the locally coupled electromechanical interface could be exploited to recognize the complex combination of low-force and high-force gestures simultaneously. For instance, the excitation–contraction signatures during the "resist and grip" gesture (Supplementary Fig. 19) could identify the release and performance of minGrip (grip with minimal forces applied) of the dumbbell rod (weight of 4 kg, rod diameter of 4.3 cm), while resisting the weight over the hand palm. Although sEMG amplitude in subject 5 showed little change in step IV and V, the drop and increase of the mechanical strain could still be clearly resolved (Supplementary Fig. 19b). Given the lower max grip force of subject 2 (22.5 kg), relatively higher sEMG signals were detected, which could better indicate the "resisting" component of the complex gestures. Although their sEMG was different, the mechanical strain patterns of both subjects appeared similar. This further validated that the identification of the electromechanical coupling can be advantageous in gesture categorization over those unimodal methods.

**Dexterity and robustness in identifying hand gestures**. In addition to recognizing gesture strength (i.e., force levels), the dexterity and robustness of the locally coupled electromechanical interface were examined with regards to its responsiveness to single-finger gestures, gesture speed, and muscle fatigue. The four tendons of FDS muscle are attached to the middle phalanges of the index, middle, ring, and little fingers. With *CoupOn* (hybrid 2

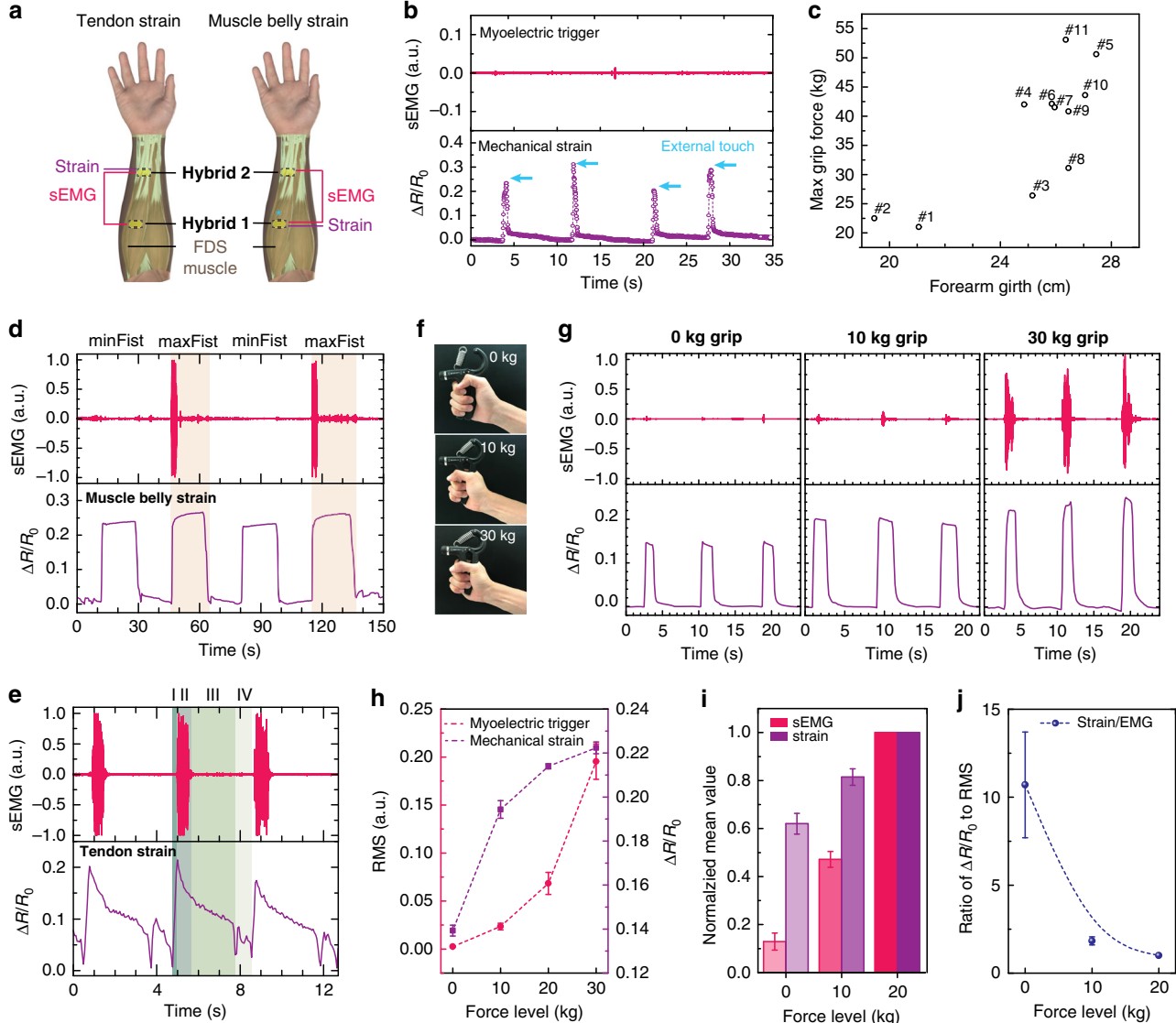

**Fig. 4 Identification of muscular excitation–contraction signatures. a** Schematic of a pair of *CoupOn* (hybrid 1 and 2) along flexor digitorum superficialis (FDS) muscle, with hybrid 1 at the FDS belly as the muscle belly strain sensor or hybrid 2 at the FDS origin as the tendon strain sensor. **b** Graph shows that resistive strain sensing of *CoupOn* incurred little influence on sEMG recording. In the case of passive external touch, the strain was detected without sEMG artifacts. **c** Circle plot shows biometrics of 11 subjects. **d** Plots show the local coupling of sEMG recording with resistive strain sensing that distinguished a minFist (minimal voluntary contraction) from a maxFist (maximal voluntary contraction). Representative of five tests. **e** Plots show the electromechanical coupling during the maxFist using the tendon strain sensor. In contrast to **d**, the excitation–contraction signatures comprised four distinct phases as follows: (I) Fist closing, (II) Fist tightening, (III) Fist relaxing, (IV) Fist opening. Data obtained on subject 9. **f, g** Photographs and plots showing the excitation–contraction signatures of hand grips at force levels of 0 kg, 10 kg, and 30 kg, respectively. **h** Plots show that both root mean square (RMS) of sEMG signals and resistive strain sensing increased with force levels. Strain sensing appeared more sensitive to the force increase in the lower end, while sEMG seemed more responsive to the force increase in the higher end. Data obtained on subject 5. **i** Bar chart shows the normalized mean RMS of sEMG and resistive strain signals. **j** Plot shows the ratio of normalized resistive strain signals to sEMG RMS. It suggests the local coupling efficiency of myoelectric trigger and mechanical strain decreases with the force level. Data obtained from the 11 subjects and normalized using respective signal amplitudes at 20 kg grips. Data are presented as mean ± SEM.

in Fig. 4a) as the tendon strain sensor, it responded differentially to the motion of different fingers against a finger exerciser (Fig. 5a, b). Among the four fingers, index flexion (subject 9) caused relatively weak sEMG peaks (amplitude ~0.02), whereas the flexion of middle and ring fingers induced slightly stronger sEMG peaks (amplitude about 0.03–0.05). On the other hand, index flexion showed moderate strain ($\Delta R/R_0 \sim 0.02$) with simple curve envelopes, whereas the middle flexion exhibited a higher resistivity increase ($\Delta R/R_0 \sim 0.05$) with two sharp peaks and the ring flexion caused a sharp resistivity decrease ($\Delta R/R_0$ about −0.06) with a broad band followed by a sharp inverse peak. By

contrast, thumb extension caused little sEMG signals and the lowest strain ($\Delta R/R_0$ about −0.01), as it was not driven by FDS muscle. This difference identified by *CoupOn* was actually consistent with the fact that tendons for the ring and middle fingers are superficial to that of the index and little fingers (equally observed in subject 2; Supplementary Fig. 20).

Although the amplitude of sEMG is commonly correlated with the level of force generation, the correlation can be complicated, as the varying contraction speed can also cause a difference in sEMG amplitude and frequency[53]. In 20 kg grips (Fig. 5c–e), slow grips (above 2 s) generated relatively weaker sEMG signals (root

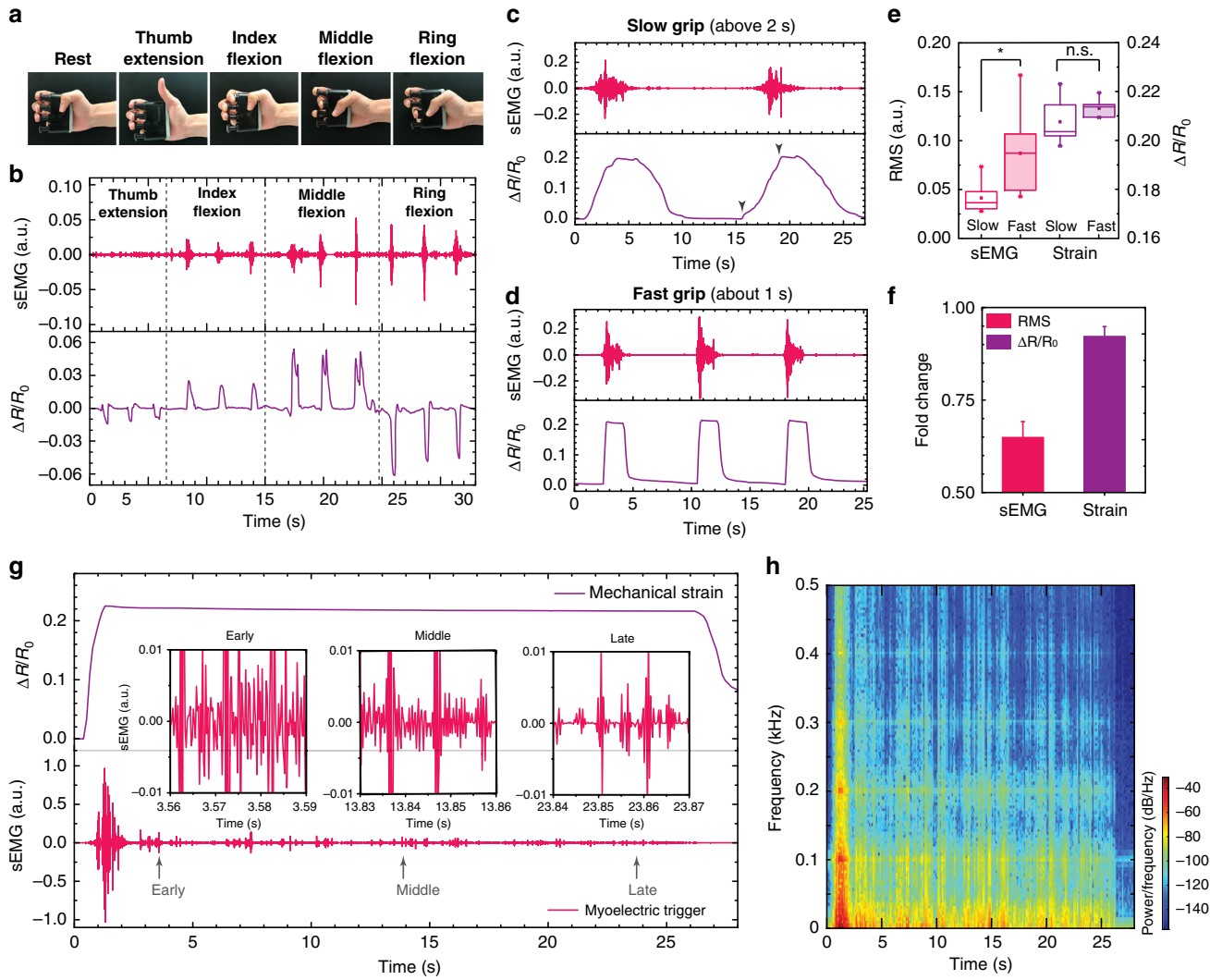

**Fig. 5 Robust recognition of finger flexions and the speed and fatigue of hand grips. a** Photographs of different finger gestures using a finger exerciser. **b** Graphs show distinct signatures of different finger flexion with the tendon strain-sensing *CoupOn* placed on the origin of FDS. Data obtained from subject 9, representative of four tests. **c**, **d** Graphs show signatures of 20 kg grips at different speeds (fast and slow). Data obtained on subject 5, representative of two tests. **e** Box chart shows the RMS of sEMG and resistive strain signals in fast and slow grips. Slow grips show significantly decaying sEMG but remained strain levels, in addition to the different waveform of sEMG and slope of strain curves. **f** Bar chart shows the relative fold change of sEMG and strain signals during slow grips against fast grips. Data obtained from seven subjects. **g** Plots of muscular excitation–contraction signatures during muscle fatiguing. Subject 5 holds the 30 kg grip with a dynamometer till failure to keep the gesture. FDS belly strain sensing is retrieved. Insets are plots of cropped 30 ms windows sEMG signals during the early, middle, and late stage of the fatiguing process. A decay in both the frequency and peak amplitude can be identified. Representative of four tests. **h** Color-encoded image shows the shift of the sEMG signal towards low frequency in **b**. A fast and dramatic decay in high-frequency signals was found in the early stage (within 5 s), followed by a gradual shift towards even lower frequency signals. Data are presented as mean ± SEM. *$P \leq 0.05$ (Student's *t*-test).

mean square (RMS) ~0.04), compared with the relatively stronger sEMG signals (RMS ~ 0.08) fast grips (within 1 s). The difference arises from that more large motor units are recruited as the contraction speed increases. Compared with fast grips, the relative fold change in sEMG was found to be ~0.65 in slow grips (Fig. 5f). By contrast, the plateau value of the mechanical strain showed no significant decrease during slow grips, but the curve envelopes revealed such information as the starting point and gesture kinetics of the fist closure. It suggests that the plateau value of the mechanical strain is relatively consistent with the grip forces, while strain curve envelopes can be indicative of the gesture speed.

Muscle fatigue leads to the decay in sEMG amplitude and frequency even before the obvious drop of contraction forces[50], probably compromising the usage of sEMG signals for active

prosthetic control. Muscle fatigue may be caused by the nerve's failure to maintain a high-frequency signal (known as nervous fatigue), or by the muscle cells with substrate shortage and metabolites accumulation (known as metabolic fatigue). It is necessary to distinguish between muscle fatigue and the change of voluntary force level during a task. During a continuous 30 kg grip (over 25 s), the excitation–contraction signature was characterized by an obvious decay in sEMG but the maintained plateau in mechanical strain (Fig. 5g). Although the nervous fatigue was indicated by the drop in sEMG amplitude (Supplementary Fig. 21) and the shift in frequency (Fig. 5h), no obvious metabolic fatigue appeared as shown by the relatively constant mechanical strain (Supplementary Movie 1). During the prolonged and tougher fatiguing task (40 kg "grip till fail"), subject 5 failed in maintaining the grip due to the metabolic

fatigue (Supplementary Fig. 22). Such metabolic fatigue was manifested by the gradual drop of the mechanical strain in all the three consecutive grips. Prior to the third consecutive metabolic fatigue, muscle vibration was captured, which showed a relatively strong sEMG spikes and multiple small spikes in the mechanical strain (Supplementary Fig. 22d). In the failure phase, the mechanical strain gradually dropped by ~57% and sEMG showed a shift towards the low frequency end (Supplementary Fig. 22e). During the consecutive 20 kg grips (~89% maximal voluntary contraction), subject 2 showed even stronger sEMG signals when the mechanical strain began to drop (Supplementary Fig. 23). This might show the failure of the muscle cells to function though the motor neurons were still trying to recruit them. Unlike previously adopted cognition implication to avoid the influence of muscle fatigue on active prosthetic control, here we found that nervous fatigue indicated by sEMG occurred ahead of the perception by the subjects, but followed by the metabolic fatigue. Therefore, the locally coupled electromechanical interface can simultaneously indicate the gesture failure and muscle fatigue types, providing a correction measure for judging muscle fatigue. More interestingly, muscular congestion was even revealed with the mechanical strain plateau decreased by ~ 36% from ~0.22 for the first grip to ~0.14 for the subsequent two grips (Supplementary Fig. 22). Such manifests of muscular congestion could potentially be used to indicate fatigue history.

**Orthogonal recognition of hand gestures for dynamic bionic manipulation**. Lastly, we demonstrated that the identified excitation–contraction signatures could orthogonally recognize the speed and strength of hand gestures, therefore be further translated to manipulate a commercial robotic hand model (called uHand) for recapitulating dynamics of the corresponding hand gestures. In our demonstration, hand grips were dissected into two steps, namely the minFist step and the tightening step (Fig. 6a). The minFist corresponded to the wrapping of a hand around an object with minimal forces applied, while the tightening step involved the force exertion for grabbing or lifting the object. By varying the speed in each step, four permutations were defined that combined slow or fast minFist with slow or fast tightening. After extracting excitation–contraction signatures of the four permutations, commands based on the aforementioned orthogonal recognition were sent to a robotic uHand for recapitulating corresponding dynamics of human hand grips. As shown in Fig. 6b, in the minFist step, the slope of resistivity change caused by slow closure was apparently lower than that by fast closure, although little sEMG signals were detected. In the tightening step, sEMG signals generated in fast tightening was significantly stronger than that in slow tightening. In addition, the amplitude of myoelectric signals attenuated due to nervous fatigue without the mechanical manifestation of metabolic fatigue, while the hand grip remained constant. By contrast, discerning slow and fast tightening by resistivity change seemed relatively challenging, since the resistivity increase from minFist step to tightening step was relatively small. The orthogonal recognition of gesture strength and speed was therefore achieved by the correlative analysis of the excitation–contraction signatures extracted in the four permutations of slow/fast minFist and slow/fast tightening.

To demonstrate the robotic manipulation, a soft ball was placed in the palm of the uHand, the ring finger of which was connected to a force gauge apparatus (Fig. 6c). Upon the command received from the coupled patterns of myoelectric triggers and mechanical strain, uHand responded by a two-step-grip fashion (Supplementary Movies 2–5). The herein generated forces appeared consistent with the force generation patterns in corresponding human grips (Fig. 6d). Moreover, the kymographs of the selected area (Supplementary Fig. 24) also revealed distinct details of gesture dynamic. Similar to human grips, forces were only applied on the soft ball in the tightening step, as evident from the projected ball diameter ($d_2 > d_1 = d_0$), as well as the grip force gauging (Fig. 6d).

## Discussion

Prior to this work, experimental approaches[19,54] and theoretical simulations[1,55] have already listed several factors that contribute to the spatiotemporal differences of the myoelectric triggers and mechanical responses. These factors include such electrochemical aspects as synaptic transmission, the propagation of action potential along myocytes and the excitation–contraction coupling, as well as the mechanical aspects like the muscle force transmission along the series elastic components and the intrinsic total deformation gradient[19,55]. This intrinsic difference in the patterns of myoelectric triggers and mechanical responses endows current methods relying on either one with a lower fidelity and inferior precision in evaluating dynamic neuromuscular performance. For instance, active prosthetic hands usually require sEMG signals at a high contraction level or from multiple channels, while precise control over single fingers can be challenging without data-driven intention decision algorithms[51]. Moreover, the fidelity of these sEMG-based methods can be seriously challenged by the prominent heterogeneity among wearers with regards to muscle shape, power, the efficiency of the electromechanical coupling (i.e., the ratio of twitch force to sEMG levels), and resistance to muscle fatigue. On the other hand, current techniques based on strain sensors are less preferable for real-time prosthetic control due to the electromechanical delay[54]. Techniques based on strain sensory arrays can be further limited by their low-sensitivity to hand gestures[56], incapability of indicating the force level[57], and being less friendly to amputees.

In this work, we have presented the cytoadhesion-inspired hybrids (so-called *CoupOn*) with locally coupled electromechanical interfaces. The advantages include the applicability of (1) in situ, (2) continuous and dynamic evaluation of voluntary muscle contraction, and (3) local electromechanical coupling of myoelectric triggers and mechanical strains. *CoupOn* is comprised of the mechanically integrated elastomer, stretchable metallic nanofilms, and adhesive ionic gel. The strong interlayer adhesion is achieved by combining the tough adhesion at the gel/metallic film interface and the cytoadhesion-inspired contact splitting via "adhesion plaques." Given the conformal adhesion and low contact impedance with skin, *CoupOn* showed supreme retrieval of electrophysiology in such relatively weak sEMG of small motor units (i.e., FDI). Meanwhile, resistivity change in the double metallic nanofilm could be exploited to track the strain of superficial skins, with a high GF of ~34 but negligible influence in skin contact impedance.

By identifying the excitation–contraction signatures of forearm muscle contraction, *CoupOn* has demonstrated the capability of orthogonally and distantly recognizing distinct hand gestures with regards to the amplitude, strength, and speed. This capability was enabled by the locally coupled electromechanical patterns of voluntary muscle contraction and could further inspire new fusion and classification algorithms towards online and accurate gesture classification[52,58–60]. By contrast, current efforts of decoding the excitation–contraction coupling process, however, usually involve arrays of multiple types of sensors[52,61,62] and bulky equipment, but fail to deliver local electromechanical coupling, rendering the continuous point-of-care monitoring less accessible. Tracking the flexion of fingers by placing sensors right on them shows improved recognition, but is not friendly to the

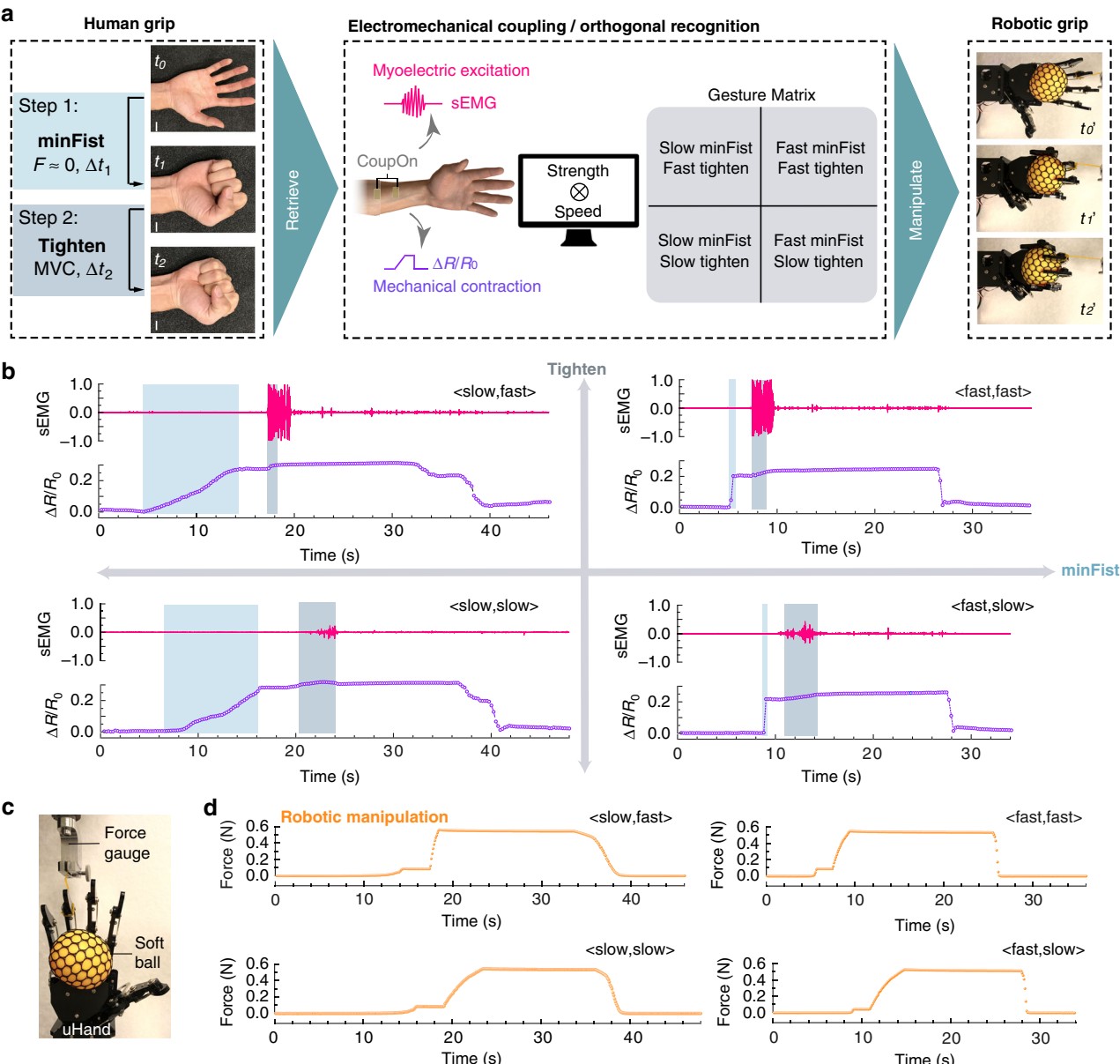

**Fig. 6 Orthogonal recognition of human grips for dynamic robotic manipulation. a** Schematic shows the process of identifying human grip signatures for manipulating robotic grips. Both human and robotic grips involve two steps, namely the minFist and tightening. The coupled electromechanical patterns retrieved by *CoupOn* can orthogonally recognize the speed and strength of human grips, therefore allowing the categorization of four permutations in the gesture matrix, according to speed of the two steps. Scale bar, 2 cm. **b** Plots show the retrieved excitation–contraction signatures corresponding to the four permutations in gesture matrix. Data obtained on subject 5. **c** Photograph shows the setup for the robotic grip of a soft yellow ball with the ring finger connected with a force gauge. **d** Plots indicate the grip force exerted on the soft ball during the four permutations of recapitulated grips.

individuals with a wrist disarticulation amputation[56,63]. In addition, those using ergometer to calibrate global force generation fail to take into consideration of the spatial complexity[20] (e.g., between the muscle belly and myotendinous junction[19]), which also refrain the subject from free movement. Other methods can involve complicated techniques (e.g., ultrasonography[19] and mechanomyography[54]), which need advanced expertise and frustrate the point-of-care documentation.

In conclusion, the development of locally coupled electro-mechanical interfaces for the dynamic identification of muscular excitation–contraction signatures would not only leverage the dexterity and robustness of prosthetic limbs and other cyber–human systems, but also advance the prognosis of neuro-muscular disorders. The demonstrated metal-hydrogel hybrids

are equally promising in implantables and cyborg tissues, by recruiting wet, soft and ionic hydrogels[64] (e.g., PEDOT hydrogels[65,66]) to interface with biological tissues. This would improve the local sensing and stimulation of biological tissues with regards to correlated electrical and mechanical aspects, towards the development of ultra-intimate human–machine merging.

## Methods

**Fabrication of *CoupOn* hybrids**. PDMS precursor (1 : 10) was spin-coated at 2000 r.p.m. onto 1H,1H,2H,2H-perfluorooctyltriethoxysilane-treated glass slide, to obtain thin PDMS film after curing at 60 °C overnight. On the surface of the cured PDMS film, a 40 nm microcracked Au film was physically deposited by a vacuum thermal evaporator (Nano 36, Kurt J. Lesker), followed by sputtering another layer 10 nm Ti (PVD 75, Kurt J. Lesker). Subsequently, the sample was modified by

grafting functional saline TMSPMA after the treatment of $O_2$ plasma (Femto Science). Tough hydrogel precursor with 2 M LiCl was sandwiched with the TMSPMA-treated sample and 1H,1H,2H,2H-perfluorooctyltriethoxy-silane-treated coverslip to form the ionotronic hybrid after being briefly vacuumed and then heated at 50 °C in a humid box overnight.

**Resistive strain sensing**. For the cyclic strain test, all tests were performed under 100% relative humidity and the sample sizes were 2 cm × 1.5 cm. One hundred percent strain was applied using a mechanical tester (C42, MTS Systems Corporation) at the speed of 2 cm/min, while the resistance was recorded by a semiconductor parameter analyzer (Keithley 4200-SCS, Tektronix). To measure mechanical strains during isometric voluntary muscle contraction, the resistivity change of the *CoupOn* hybrid adhered to the subjects' forearm was recorded by the semiconductor parameter analyzer (Keithley 4200-SCS, Tektronix).

**sEMG recording and signal processing**. sEMG signals of the FDI test (opposition of the thumb and index finger), standardized hand grip at different force levels and single-finger flexions were recorded by a home-customized toolkit. The toolkit includes an amplifier and a 50/60 Hz filter. The obtained raw sEMG signals were filtered with Butterworth low pass filter (5–500 Hz) and rectified for RMS (window size 100 ms) and time-frequency analysis. SNR analysis was performed with home-customized codes in Matlab obtained through the equation intensity and background noise level.

**Subjects and tasks**. Quantitative validation of *CoupOn* hybrids were conducted on 11 subjects (3 female and 8 male; Supplementary Table 1): aged 21–32 years, body mass index of 17.5-24.7, maximal forearm girth of 19.5–27.5 cm, and maximal grip forces of 21.0–53.1 kg (measured with an electronic hand dynamometer Camry, EH101). The protocol of this study is approved by NTU Institutional Review Board. Subjects kept the forearm in a comfortable position while resting on the table with the elbow at an angle of ~90°, avoiding wrist flexion, extension, deviation, pronation, and supination. Subjects used the grip dynamometer to perform grips at the force level of 0, 10, 20, and 30 kg (30 kg grips for male subjects only), within 1 s. Slow grips (above 1.5 s) of 20 kg were performed at subjects' will. To avoid muscle fatigue during these tasks, subjects only performed five to eight times during each set and repeated two to four sets with a resting interval (1–2 min) in between, after proper gesture training. Meanwhile, muscle fatiguing was recorded by holding the 30 kg grip over ~25 seconds. Subjects also conducted the "grip till fail" fatigue tasks by holding 20 kg or 40 kg grips till the physical failure and gradual release of the grips for two to three consecutive times with 1 min intervals. In the "resist and grip" gesture, the subject resisted a 4 kg dumbbell on hand palm while performing the minGrip (grip the dumbbell rod with minimal forces applied). Standardized single-finger gestures were performed with a finger exerciser (Flanger, FA-10P), including the extension of the thumb and the flexion of the index, middle and ring fingers, independently.

**Reporting summary**. Further information on research design is available in the Nature Research Reporting Summary linked to this article.

## Data availability
Data that support the findings of this study are available from the corresponding author upon a reasonable request.

## Code availability
The customized code for sEMG analysis is available from the corresponding author upon a reasonable request.

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

## Acknowledgements

The project was supported by the Agency for Science, Technology and Research (A*STAR) under its AME Programmatic Funding Scheme (A18A1b0045), the National Research Foundation (NRF), Prime Minister's office, Singapore, under its NRF Investigatorship (NRF-NRFI2017-07), and Singapore Ministry of Education Tier 2 (MOE2017-T2-2-107). N.M. was supported by Japan Society for the Promotion of Science Overseas Research Fellowships.

## Author contributions

X.C. and P.C. conceptualized the project. X.C. and P.C. designed the experiments and optimized the method. X.C., P.C., C.W., L.P., N.M., J.W., and G.C. wrote the paper. P.C., C.W., L.P., K.H., Z.C., C.L., W.Z., J.Y., M.W., and Y.J. performed the experiments and analyzed the data.

## Competing interests

The authors declare no competing interests.
