## [Peer Review File · Nature Communications]

Reviewers' comments:

Reviewer #1 (Remarks to the Author):

This is certainly nice paper that can fully activate imaginations and hopes of well-designed materials for actual and realistic usages. Good points of this manuscript is (i) lots of good demonstrations and (ii) high-level manuscript organization. Publication of this type of research work in well-authorized public media such as Nat. Commun. would have good impacts about important roles of science and technology to social media. I basically recommend publication of this work. However, this manuscript lacks several fundamental scientific data (although advanced points for demonstrations are well satisfied). Please see below.

1) One of the important keys in the used good materials is molecularly engineered interface illustrated in Figure S3. However, this structures were not experimentally proved. Such structures and presence of covalent bonding are usually proved by appropriate spectroscopies such as IR and/or NMR. Without these data, the used materials systems cannot be scientifically proved. This point has to be fixed.

2) Although surface morphologies were supplied by precise observation of SEM and AFM, more fundamental aspects such as flat natures of the materials were not proved by low-resolution images (cross-sectional images etc). Please supply such fundamental data.

3) As information source of recent progresses in science and technology for interfaces between bio (human) and devices, well-described review by Nishizawa (see, Bull. Chem. Soc. Jpn. 91, 1141-1149 (2018)) had better be considered.

Reviewer #2 (Remarks to the Author):

The manuscript describes an electromechanical interface that detects skin deformations concurrently with surface EMG signals. The stated aim of this interface is to improve decoding of motor intention in man-machine interfacing systems.

The sensor development is interesting and well detailed. The need for a coupled mechanical and EMG sensor is justified and this development would likely have some relevant applications. Nonetheless, the manuscript suffers from rather poor writing style and, especially, from a very limited test of the proposed device. The test of the device is done by recording activity of forearm muscles during fist clenching. There is no information on the number of subjects studied in the experiments nor on the way in which force was measured. The authors vaguely report results on 'loose' and 'tight' fist clenching, without specifying how these two tasks are objectively defined. In this way, it is impossible to repeat these experiments. Also, the results of these tests are very peculiar since the authors report no EMG during the loose fist clenching while of course muscle activity is needed for such task.

Another test that the authors report is related to classifying four classes for robotic control. Again, the task is fist clenching and this time it is divided into four sub-tasks corresponding to slow/fast closure and tightening. The reason for considering these classes is unclear since closure and tightening would be the same command to a robotic hand (it would be closure without object interaction and tightening with object interaction but clearly the two actions would require only one command from a man-machine interface). Moreover, again, the number of subjects of these tests is not reported, nor the exact experimental conditions. Also, the way the four classes are classified is not described.

For both experimental tests, the authors emphasize that only one of the two measurements (EMG or mechanical deformation) would not be sufficient to achieve the performance shown. However, this statement is purely qualitative and not supported by any quantitative evaluation. There are extremely accurate EMG systems for proportional control of one degree of freedom that work much better than shown in this manuscript without the need for additional signals (see extensive literature on active prosthesis control).

Overall, the proposed tests are not rigorous and do not provide any quantitative demonstration of the potential of the technique. The manuscript therefore is limited to the technical development of an interesting sensor but lacks any evidence of its applicability in man-machine interfacing.

Reviewer #3 (Remarks to the Author):

The manuscript "Locally-coupled electromechanical interfaces based on cytoadhesion-inspired hybrids to identify muscular excitation-contraction signatures" by Cai et al. describes a novel design of combined strain and myoelectric sensor in form of a wearable patch. It is demonstrated, that this kind of sensor can work in identification of different hand gestures, in particular to distinguish between movements that could not be distinguished with either strain or myoelectric sensors alone. The make up of the patch especially solves previous problems in the field by combining conformity and stability of metal films and gels. In particular, a bio-inspired form of "focal adhesion" of the gel film by holes and cracks in the gold film and subsequent silanization is presented resulting in excellent performance of the sensor.

The manuscript is overall well written and presents a very interesting approach to the combined strain and myoelectrical sensing. I suggest acceptance of the manuscript with a few minor suggestions to be implemented by the discretion of the authors:

- how are the holes in the gold film formed? The authors give some vague reference to the crack formation, but is a mechanism for the (in average) quite regular and quite uniformly sized holes known? This should be discussed at the manufacturing description.

- can the mean minimum distance / size of the holes be influenced by the manufacturing parameters and does this affect the stability of the gel / metal film?

- did the authors do a comparison on adhesion directly on a film with and without these holes? So what do they give in addition to the cracks alone and to a flat gold film?

Some minor language points:

- please check on use of articles, e.g. in the abstract "we established the locally-coupled electromechanical interface" should be "a locally-coupled" and there are several additional occasions throughout the manuscript.

- on page two "surface electrogram (sEMG) electrodes", actually the abbreviation sEMG belongs to "surface electromyogram" or "surface electromyography", which is of course a type of electrogram / electrography, but when introducing the specific abbreviation it should be also specific.

- the name "uHand" is used without any definition or short description or reference. I don't think that this is a generally known proper name, so I would suggest to give it a short description e.g. "a robotic hand model, called uHand" with a matching reference or similar on first occasion.

Detailed Responses to Referees' Comments

We would like to appreciate the three reviewers for their constructive suggestions and insightful comments. Their suggestions have improved this manuscript, in terms of clarity and readability, as well as the rigorous and quantitative aspects of the experiments. The point-to-point responses to the reviewers' comments are detailed below.

Reviewer #1:

This is certainly nice paper that can fully activate imaginations and hopes of well-designed materials for actual and realistic usages. Good points of this manuscript is (i) lots of good demonstrations and (ii) high-level manuscript organization. Publication of this type of research work in well-authorized public media such as Nat. Commun. would have good impacts about important roles of science and technology to social media. I basically recommend publication of this work. However, this manuscript lacks several fundamental scientific data (although advanced points for demonstrations are well satisfied). Please see below.

Response: We appreciate the positive and encouraging comments by the reviewer, and we are equally inspired by the kind suggestions from the reviewer. The suggestions have improved the quality of the manuscript with respects to the chemical and morphological characterization of the electrode interface.

1) One of the important keys in the used good materials is molecularly engineered interface illustrated in Figure S3. However, this structures were not experimentally proved. Such structures and presence of covalent bonding are usually proved by appropriate spectroscopies such as IR and/or NMR. Without these data, the used materials systems cannot be scientifically proved. This point has to be fixed.

Response: The reviewer's suggestion on investigating the molecular composition and bond formation could indeed improve readers' understanding of the interface engineering. Therefore, we exploited confocal Raman spectroscopy (Supplementary Fig. S4) and FTIR (Supplementary Fig. S5), and have confirmed the salinization of TMSPMA and subsequent anchorage of tough gel. Detailed interpretation of the spectrum and characteristic absorption peaks/bands are included in the figure caption. In addition, it is noteworthy that the characteristic absorption peaks for TMSPMA seemed to be attenuated when grafted onto Ti surfaces (Fig. R1). The spectra of TMSPMA (drop-casted on KBr pellet) exhibited strong absorption peaks at 1717 cm^{-1} for the stretching vibration of the C=O group and 1635 cm^{-1} for the vinyl group, which remarkably attenuated in the spectra of *Thick TMSPMA on Ti/Au@elastomer* and even disappeared in that of *Thin TMSPMA on Ti/Au@elastomer*. Here, *Thick TMSPMA on Ti/Au@elastomer* sample is prepared by drop-casting 200 μL TMSPMA solution (2 wt.% in water, pH 3.5) on plasma-treated Ti surfaces and dried in oven at 50 $^{\circ}\text{C}$ without ethanol rinse, *Thin TMSPMA on Ti/Au@elastomer* sample is prepared by allowing the condensation reaction at room temperature and rinsed with ethanol thoroughly. By contrast, the emerging strong peak at 1007 cm^{-1} in both spectra indicates the formation of Si-O-Ti bridges, and therefore the grafting of TMSPMA onto Ti surfaces.

Figure R1. Graph showed the FTIR spectra of TMSPPMA on different surfaces.

2) *Although surface morphologies were supplied by precise observation of SEM and AFM, more fundamental aspects such as flat natures of the materials were not proved by low-resolution images (cross-sectional images etc). Please supply such fundamental data.*

Response: We appreciate the reviewer for suggesting the cross-sectional characterization of the *Coup-On*. To confirm the flat nature of *Coup-On*, we have first investigated the cross-section SEM image of the whole patch at low magnification (350 \times) and observed the intimate contact of the tough gel and the Au/Ti@elastomer (Supplementary Fig. S10a). To better resolve the flatness of the interface between the metallic nanofilm (Au/Ti) and gel (Supplementary Fig. S10b), we removed the elastomer layer and confirmed the flatness of the hybrid and the consistent thickness. Interestingly, the intimate contact between the gel and metallic nanofilm was equally observed, implying the tough bonding formation at the interface.

Figure S10b. Cross-sectional SEM image of the Coup-On patch with the elastomer removed after freeze-drying. *White arrows* in the inset showed the intimate contact between the tough gel (*grey color*) and the metallic nanofilm (*magenta color*), implying the formation of the chemical bond at their interface.

3) *As information source of recent progresses in science and technology for interfaces between bio (human) and devices, well-described review by Nishizawa (see, Bull. Chem. Soc. Jpn. 91, 1141-1149 (2018)) had better be considered.*

Response: We thank the reviewer for the advice, and we have further included a brief discussion on recent advances at the interface between human/biology and devices in the discussion session of the revised manuscript. (Pg. 16, highlighted in yellow), and highlighted the prospects of introducing soft, wet and ionic hydrogels at the interface.

Reviewer #2:

The manuscript describes an electromechanical interface that detects skin deformations concurrently with surface EMG signals. The stated aim of this interface is to improve decoding of motor intention in man-machine interfacing systems. The sensor development is interesting and well detailed. The need for a coupled mechanical and EMG sensor is justified and this development would likely have some relevant applications.

Response: We are glad that the reviewer shares the idea that the local coupling of skin deformation and surface EMG is necessary for decoding of the motor intention for man-machine interfacing systems. To further justify this, we have given additional discussions on unsolved challenges of current techniques in addressing the spatiotemporal differences in patterns of the myoelectric trigger and mechanical strains (Pg. 14 & 15, highlighted in yellow).

Nonetheless, the manuscript suffers from rather poor writing style and, especially, from a very limited test of the proposed device. The test of the device is done by recording activity of forearm muscles during fist clenching. There is no information on the number of subjects studied in the experiments nor on the way in which force was measured. The authors vaguely report results on ‘loose’ and ‘tight’ fist clenching, without specifying how these two tasks are objectively defined. In this way, it is impossible to repeat these experiments.

Response: We thank the reviewer for inspiring advice. To improve clarity and repeatability of the experimental conditions, we have further conducted the tests on 11 subjects with varying forearm girth and maximal grip forces (Fig. R2; Supplementary Table S1). To exclude such vagueness, we re-define the loose and tight fist clenching as the fist closure with minimal voluntary contraction (named as minFist) and the fist-clenching with maximal voluntary contraction (named as maxFist), respectively. To improve the repeatability of our tests, we standardized the grip posture and grip force levels (0 kg, 10 kg, 20 kg and 30 kg) using a dynamometer. In order to ensure the gesture amplitude, 0 kg grips are performed by removing the spring in the dynamometer and keep it at the “closed” position. During the tasks, subjects were asked to keep the forearm in a comfortable position while resting on the table with the elbow at an angle of approximately 90 degrees, avoiding wrist flexion, extension, deviation, pronation, and supination (see Experimental Methods in Supplementary Information for more details). In addition, a finger exerciser is also used to standardize the gestures of finger flexion.

Figure R2. **a**, Circle plot shows the max grip force against forearm girth of 11 subjects. **b**, Photographs of hand grips using a grip dynamometer at different force levels of 0 kg, 10 kg, 20 kg, and 30 kg. **c**, Photographs of different finger gestures using a finger exerciser.

Also, the results of these tests are very peculiar since the authors report no EMG during the loose fist clenching while of course muscle activity is needed for such task.

Response: We thank the reviewer for stating their confusion on no sEMG signals detected for “loose fist-clenching” (similar to “0 kg grip” in the revised manuscript). We know that “loose fist-clenching” (“minFist” or “0 kg grip” in the revised manuscript) involves the flexion of distal interphalangeal joints of fingers that is driven by flexor digitorum profundus (FDP) in the deep layer. In the meanwhile, motor units can be classified into type I (low twitch threshold), type IIa (high twitch threshold, fatigue-resistant), type IIb (high twitch threshold, fast-fatigue). In the gestures of “loose fist-clenching” (“minFist” or “0 kg grip” in the revised manuscript), type I motor unit would be firstly recruited since no resisting forces were exerted by the grip dynamometer. In addition, the body tissues can act as a filter against low-frequency EMG signals, and the FDS muscle is in the intermediate layer of the forearm, hence detection of such low surface EMG signals could be challenging and highly dependent on neuromuscular conditions of the subjects. Actually, we did observe a few subjects (*e.g.*, subject 5 in Fig. 4g) showing a low level of observable sEMG signals in 0 kg grips. This also inspires us to develop further improved skin-mountable electromechanical interfaces towards detecting the activation of type I motor units at little or low twitch forces.

Another test that the authors report is related to classifying four classes for robotic control. Again, the task is fist clenching and this time it is divided into four sub-tasks corresponding to slow/fast closure and tightening. The reason for considering these classes is unclear since closure and tightening would be the same command to a robotic hand (it would be closure without object interaction and tightening with object interaction but clearly the two actions would require only one command from a man-machine interface). Moreover, again, the number of subjects of these tests is not reported, nor the exact experimental conditions. Also, the way the four classes are classified is not described.

Response: We sincerely appreciate the reviewer for the critical evaluation of our demonstration of implementing the locally-coupled electromechanical interface (*Coup-On*) in manipulating a robotic hand. Firstly, we dissect the fist-clenching (grip gesture) into two steps, namely the minFist step and the tightening step (Fig. 6a). The minFist corresponds to hand wrapping around an object, while the tightening step involved the force exertion for grabbing or lifting the object. In the minFist step, hand contact with the object is made but little forces are exerted on the object. While it might seem identical for the minFist closure and tightening step grabbing a soft object since low resisting forces are counteracted against the hand in the tightening step. However, the situation can be quite different when grabbing a rigid object that is highly resistant to deformation. In this case, the tightening step involves the increased exertion of grip forces but little change of the finger position and fist shape. Unfortunately, we have no access to a robotic hand with the hydraulic force control unit to demonstrate this. Instead, we used the deformation of a soft ball to indicate the exertion of grip forces on it. In Fig. 6e, the projected diameter of the ball ($d_2 > d_1 = d_0$) in the tightening step is larger than that in the minFist step (no forces on the ball) and before grip execution (no contact made).

In the revised manuscript, we further justified that the advantages of *Coup-On* in identifying the two aspects of the gesture (gesture speed and strength), through the tasks of varying gesture strength (*i.e.*, grips a different force level), and gesture kinetics (*i.e.*, fast vs. slow grip at the identical force level). To demonstrate the potential application of such capabilities, we therefore vary the speed in the minFist and tightening step, and define four permutations that combined slow or fast minFist with slow or fast tightening. Then the excitation-contraction signatures in the four permutations of subject 5 during the maximal voluntary contraction were retrieved to manipulate a robotic hand. Only one subject was tested since we found that the excitation-contraction signatures of different subjects can be quite

personalized and the algorithm needs to be re-calibrated. However, one wearer only needs one prosthetic hand and our focus is not on developing a universal algorithm.

For both experimental tests, the authors emphasize that only one of the two measurements (EMG or mechanical deformation) would not be sufficient to achieve the performance shown. However, this statement is purely qualitative and not supported by any quantitative evaluation. There are extremely accurate EMG systems for proportional control of one degree of freedom that work much better than shown in this manuscript without the need for additional signals (see extensive literature on active prosthesis control). Overall, the proposed tests are not rigorous and do not provide any quantitative demonstration of the potential of the technique. The manuscript therefore is limited to the technical development of an interesting sensor but lacks any evidence of its applicability in man-machine interfacing.

Response: We appreciate the inspiring comments raised by the reviewer. We have referred to literature on active prosthesis control and found that there are three important pillars in such human-machine interfaces (HMI), namely the sensors to detect the necessary biological information, the algorithms to fuse signals from different sensors and infer human intentions, as well as the robotics to fulfil human commands. Although current active prosthesis control shows much progress in developing improved algorithms for EMG data interpretation and robotics with more degrees of freedom for actuation, however, the EMG electrodes are still either rigid (*e.g.*, dry metal electrodes) or non-stretchable (*e.g.*, wet Ag/AgCl electrodes). Rigid dry electrodes can suffer from a low SNR ratio and artifacts caused by displacement and pre-press for skin contact. Non-stretchable wet electrodes also deliver sEMG signals inferior to our *Coup-On*, which can be worsened by the stretch-induced interlayer delamination (Fig. 3e-g). This manuscript focuses on the locally-coupled bimodal sensors since the dextrous control of robotic hands can be highly advanced by the precise dissection of human hand gestures with regards to amplitude, strength, and kinetics. In addition to the potential application in HMI, the presented locally-coupled bimodal sensor may also be used for the prognosis of neuromuscular diseases (*e.g.*, Parkinson's diseases), since it is shown to retrieve FDI sEMG signals superior to commercial *Vitrode* (Fig. 3h-j).

In the revised manuscript, we have further quantified the coupling efficiency of the myoelectric trigger and mechanical contraction. We have recruited 11 subjects to perform standardized tasks with both a finger exerciser and a grip dynamometer at different force levels. It reveals some interesting characteristics of our locally-coupled electromechanical interfaces: 1) Strain sensing is good at resolving gestures at a lower force level, whereas sEMG recording is more responsive to gestures at a higher force level; 2) the coupled signatures of the myoelectric trigger and mechanical strain can recognize flexions of single fingers, contraction strength and speed, and muscle fatiguing.

Reviewer #3:

The manuscript "Locally-coupled electromechanical interfaces based on cytoadhesion inspired hybrids to identify muscular excitation-contraction signatures" by Cai et al. describes a novel design of combined strain and myoelectric sensor in form of a wearable patch. It is demonstrated, that this kind of sensor can work in identification of different hand gestures, in particular to distinguish between movements that could not be distinguished with either strain or myoelectric sensors alone. The make-up of the patch especially solves previous problems in the field by combining conformity and stability of metal films and gels. In particular, a bio-inspired form of "focal adhesion" of the gel film by holes and cracks in the gold film and subsequent silanization is presented resulting in excellent performance of the sensor.

The manuscript is overall well written and presents a very interesting approach to the combined strain and myoelectric sensing. I suggest acceptance of the manuscript with a few minor suggestions to be implemented by the discretion of the authors:

Response: We are most glad to hear the positive comments by the reviewer, and we are equally inspired by the kind suggestions from the reviewer. We have also referred to the review's minor suggestions and further improved the manuscript.

- how are the holes in the gold film formed? The authors give some vague reference to the crack formation, but is a mechanism for the (in average) quite regular and quit uniformly sized holes known? This should be discussed at the manufacturing description.

- can the mean minimum distance / size of the holes be influenced by the manufacturing parameters and does this affect the stability of the gel / metal film?

Response: We thank the reviewer for the comments. To understand the possible crack formation mechanism, we further varied the elastomer rigidity, since the crack formation is not observed on Au film deposited on a stiff substrate (e.g., glass). It seemed that substrate rigidity can influence both the crack formation and the packing of Au NPs in the continuous zone (Fig. S2b), which could be attributed to the equilibrium of Au NPs adhesion and mobility on the elastomer. The increase of substrates rigidity appeared to increase the adhesion but reduce the mobility of Au NPs, as evident from the relatively continuous Au film on PDMS (1:4) substrates. On PDMS (1:7) substrates, we observed the "hole-like" structure with the small gold patch in the centre and several long cracks at the periphery. It suggests that the formation of "hole-like" structures might be initiated by the convergence of different propagating microcracks. In addition, we also promoted the adhesion of Au NPs by introducing 3 nm Cr on PDMS (1:10) and found the "hole-like" structure formation. Interestingly, the Au speckles are less smashed than those on PDMS (1:10, no Cr layer), but more smashed than those on PDMS (1:7, no Cr layer).

Since the condition for forming such "hole" structures seems stringent, we have also demonstrated the use of PVA disks as sacrificing masks through spray-coating (Fig. S2c). The masks are coated by spraying 5 % PVA solution and air-dried before Au deposition and rinsed with warm water (50 °C) after Au deposition, forming "holes" with tunable sizes.

Figure S2. SEM images show the influence of **a)** substrate rigidity and **b)** adhesion on forming microcracked Au nanofilm. The variability in substrate rigidity can influence the size and distribution of microcracks, as well as the formation of “smashed Au speckles”. The increase of substrates rigidity appears to promote the adhesion but reduce the mobility of Au NPs, as evident from the relatively continuous Au film on PDMS (1:4) substrates. On PDMS (1:7) substrates, the “hole-like” structure is shown with the small gold patch (*White arrow*) in the centre and several long cracks at the periphery. This suggests that the formation of “hole-like” structures might be initiated by the convergence of different propagating microcracks. *Black arrow* in (b) indicates the promoted adhesion (with 3 nm Cr) can impede the formation of “smashed gold particles” in the “hole”. It suggests that conditions can be stringent for forming the “hole-like” structure with smashed Au speckles, probably demanding the equilibrium between AuNPs mobility and adhesion. Scale bar: 1 μm . **c,** Schematic and SEM images show that the size of “hole-like” microstructures can be modulated when spray-coating PVA disks as the sacrifice masks. It is noteworthy that the “holes” (*Green arrow*) are free of gold deposition, which might be beneficial for penetration of tough gel, therefore, the formation of “adhesion plaques”. Scale bar: 10 μm .

- did the authors do a comparison on adhesion directly on a film with and without these holes? So what do they give in addition to the cracks alone and to a flat gold film?

Response: We thank the reviewer for the insightful comments. We have compared the gel adhesion on a flat gold film (Fig. 2g) and found that the adhesion was rather poor (below 10 N/m). This showed that the penetration of gel would be necessary for promoting interlayer adhesion.

Some minor language points:

- Please check on use of articles, e.g. in the abstract "we established the locally-coupled electromechanical interface" should be "a locally-coupled" and there are several additional occasions throughout the manuscript.

- on page two "surface electrogram (sEMG) electrodes", actually the abbreviation sEMG belongs to "surface electromyogram" or "surface electromyography", which is of course a type of electrogram / electrography, but when introducing the specific abbreviation it should be also specific.

- The name "uHand" is used without any definition or short description or reference. I don't think that this is a generally known proper name, so I would suggest to give it a short description e.g. "a robotic hand model, called uHand" with a matching reference or similar on first occasion.

Response: We are grateful to the reviewer for spotting the errors. We have corrected these errors and also proofread them throughout the manuscript. As for “uHand”, it is a robotic hand model we purchased, and we have annotated this in the revised manuscript.

Reviewers' comments:

Reviewer #1 (Remarks to the Author):

Revisions and answers are fine. The revised version becomes acceptable.

Reviewer #2 (Remarks to the Author):

Thank you for the revision and detailed replies. My main previous concern was the relatively weak human experimental part. This has improved, most notably by the inclusion of measures on 11 subjects. Nonetheless, I still have reservations on the way the results are presented and commented in the view of human-machine interface potential of the proposed technology. In the text and figures, it is very difficult to understand how the results are reported. In some cases, there are results presented for a representative subject, in others from a few subjects. I cannot find a place where a clear statistical analysis of the results from all subjects is presented. This makes it very difficult to analyse and judge on the results. The presentation of results is almost always qualitative. I cannot find clear answers to questions like "does the proposed sensor allow a classification accuracy of gestures superior to the separate use of the two measurement modalities?". This should be tested directly and statistically analysed. I have read carefully the revised manuscript a number of times looking at this information but it seems difficult to extract from the manuscript. Similarly, I cannot find clear description of the results on muscle fatigue. Figure 5 seems to report representative results from subject 5 and subject 9 but the results on the full subject sample is not reported. What is the information extracted by the proposed sensor about fatigue? From the revised text, I believe it is amplitude and frequency content of the EMG, which can already be done with classic EMG recordings (the finding of a spectral shift in EMG and a change in amplitude with fatigue is very well known since several decades). How different is the information collected by the proposed sensor on fatigue and how can this be used to compensate for fatigue in an application such as prosthetic control (this is implied in the discussion but without clear results supporting the statement)?

Overall, although I recognize the additional work done by the authors, I find the manuscript still suffering from a similar weakness in the human investigations, as I outlined in my original report.

Reviewer #3 (Remarks to the Author):

The authors have even further improved the quality of the manuscript and added interesting and valuable additional data. I recommend acceptance of the manuscript as is.

On minor comment (if the authors want still to include/answer, but it is not necessary): I think they slightly misunderstood my question idea here:

"- did the authors do a comparison on adhesion directly on a film with and without these holes? So what

do they give in addition to the cracks alone and to a flat gold film?

Response: We thank the reviewer for the insightful comments. We have compared the gel adhesion on a flat gold film (Fig. 2g) and found that the adhesion was rather poor (below 10 N/m). This showed

that the penetration of gel would be necessary for promoting interlayer adhesion."

Which was not aiming at a total smooth film but a film with cracks but no "holes". But maybe this also means that in all cracked films also the holes appear, so it might be hard to separate the influence of these phenomena (though with the new introduced spray technique, the authors could have a look at a "just holes no cracks" film to try to separate the influence.

Detailed Responses to Referees' Comments

We are grateful to the three reviewers again for their additional comments and suggestions. Their suggestions have improved this manuscript, especially in the rigorous and quantitative aspects of the human experiments. The point-to-point responses to the reviewers' comments are detailed below.

Reviewer #1 (Remarks to the Author):

Revisions and answers are fine. The revised version becomes acceptable.

Response: We appreciated very much for the positive comments on our revisions. The reviewer's suggestions in the previous report help us to improve the quality of the manuscript.

Reviewer #2 (Remarks to the Author):

Thank you for the revision and detailed replies. My main previous concern was the relatively weak human experimental part. This has improved, most notably by the inclusion of measures on 11 subjects.

Response: We appreciate the constructive comments from the reviewer. Also, we thank the recognition of additional human experiments to validate the advantages of our locally-coupled electromechanical interfaces to understand muscle contractions. We have further demonstrated other gestures to better validate such advantages.

Nonetheless, I still have reservations on the way the results are presented and commented in the view of human-machine interface potential of the proposed technology. In the text and figures, it is very difficult to understand how the results are reported. In some cases, there are results presented for a representative subject, in others from a few subjects. I cannot find a place where a clear statistical analysis of the results from all subjects is presented. This makes it very difficult to analyse and judge on the results.

Response: We thank the reviewer for pointing out the concern on statistical analysis of all subjects. The confusion arose from three aspects: 1) The subjects' maximal grip forces are different, and the grip tasks are different for subject 1-3 and subject 4-11, hence grouped statistics were shown; 2) The inter-subject variability in their electromechanical signatures and coupling efficiency seemed remarkable; 3) Representative results of several subjects were presented only, due to limited manuscript length.

In this revision, we have further included statistical analysis on the 11 subjects (Fig. 4i, j) with data normalized against respective 20 kg grips. The coupling efficiency of myoelectric triggers and the mechanical strain can vary in different gestures (Supplementary Fig. 17 & 18), and among different subjects. The coupling efficiency seems correlating with their forearm girth and max grip forces as shown in the below diagram (Supplementary Fig. 19). In addition, we have included the representative electromechanical signatures of standard grips by several more subjects (Supplementary Fig. 16).

Supplementary Figure 19 | Diagram shows the correlation of the electromechanical coupling efficiency with subjects' forearm girth and max grip forces. Data obtained from the 11 subjects during 10 kg grips. sEMG of subject 1-3 and subject 4-11 are respectively normalized against 20 kg grips and 30 kg grips. *White asterisks* locate each subject by the max grip forces and forearm girth. Color coding and fitting conducted in Origin 9.1.

The presentation of results is almost always qualitative. I cannot find clear answers to questions like "does the proposed sensor allow a classification accuracy of gestures superior to the separate use of the two measurement modalities?". This should be tested directly and statistically analysed. I have read carefully the revised manuscript a number of times looking at this information but it seems difficult to extract from the manuscript.

Response: We thank the reviewer for stating the interest in further proof of the advantages of the bimodal recording for gesture recognition. We have previously demonstrated that the EMG sensing mode is better at tracking gestures at higher force magnitude (10 to 30 kg grips), while the strain sensing mode is better at resolving forces at lower force magnitude (0 kg grips). Such divergent sensitivity to gestures could be exploited to recognize the complex combination of low-force and high-force gestures simultaneously. Accordingly, we further define the “resist and grip” gesture, in which the FDS muscle actuates the minGrip (grip with minimal forces applied) while it is resisting to a 4 kg dumbbell on the palm. The bimodal recording could therefore resolve the minGrip and its release with “resisting” the 4 kg weight as the background (Supplementary Fig. 20). This gesture represents many other complicated ones involving the force-bearing and gentle-manipulating components, such as the daily scenario that we grip light objects with fingers while holding heavy stuff with the hand.

Meanwhile, we agree that the gesture classification accuracy is important in quantifying advantages of one recognition technology. Actually, there have been a few reports demonstrating the improved classification accuracy by using multiple sensors (Ref. 60, 62, 63). And we have further shown that the mechanical strain can be quite different in different location of the FDS muscle (*e.g.*, belly and tendon) when performing the same tasks (*e.g.*, MVC and 20 kg fast grips). It shows the spatial difference in the electromechanical coupling of muscle contraction and therefore implies the necessity of such local coupling (Fig. 4d-e; Supplementary Fig. 15). It is equally emphasized the optimal fusion algorithms are critical, and many computer scientists are contributing to such algorithms based on commercial sensors (Ref. 53, 59, 61, 64;). While this manuscript focuses on developing the locally-coupled electromechanical sensor *Coup-On*, we also hope the validated advantages will inspire computer scientists to be involved in optimizing fusion algorithms for such locally-coupled multi-modal sensors. The relevant discussion is added in Page 16.

Supplementary Figure 20 | Resolving low-force and high-force components in the “resist and grip” gesture.

a, Photographs show the 6 steps in the “resist and grip” gesture. The elbow angle kept constant during the whole process. The dumbbell was 4 kg, with the rod diameter of 4.3 cm. **b-c**, Plots show the excitation-contraction signatures of subject 5 and 2 when performing the “resist and grip” gesture. Patterns of the mechanical strain of the two subjects appear similar, though their sEMG signals appear distinct. Low-force component can be resolved from the high-force gestures through analyzing the electromechanical coupling. For subject 5, minGrip release (step iv) and minGrip (step v) while maintaining the wrist flexion exhibit little distinguishable characteristics in the sEMG signal, but obvious two plateau in the mechanical strain signal. sEMG signals of subject 2 and 5 are respectively normalized with their respective signals during the standard 20 kg grip and 30 kg grip.

Similarly, I cannot find clear description of the results on muscle fatigue. Figure 5 seems to report representative results from subject 5 and subject 9 but the results on the full subject sample is not reported. What is the information extracted by the proposed sensor about fatigue? From the revised text, I believe it is amplitude and frequency content of the EMG, which can already be done with classic EMG recordings (the finding of a spectral shift in EMG and a change in amplitude with fatigue is very well known since several decades). How different is the information collected by the proposed sensor on fatigue and how can this be used to compensate for fatigue in an application such as prosthetic control (this is implied in the discussion but without clear results supporting the statement)? Overall, although I recognize the additional work done by the authors, I find the manuscript still suffering from a similar weakness in the human investigations, as I outlined in my original report.

Response: We appreciate the reviewer for comparing the conventional method of decomposing EMG and our electromechanical coupling method for understanding muscle fatigue. Indeed, conventional methods can monitor the nervous muscle fatigue by analyzing the frequency shift and amplitude reduction. However, our method could further distinguish the two types of muscle fatigue, namely the nervous fatigue (failure of maintaining the myoelectric trigger) and metabolic fatigue (failure of the myofibers, therefore maintaining the twitch forces and mechanical strain). While sEMG indicated the nervous fatigue during the 30 kg grip over 25 seconds (Fig. 5), no metabolic fatigue occurred and little change in the mechanical strain was observed. However, during the prolonged and tough fatiguing task

(40 kg grip for subject 5; 20 kg for subject 2), subjects failed to maintain the grip due to the metabolic fatigue (Supplementary Fig. 23 & 24), which was clearly captured by the gradual drop of mechanical strain. In addition, the coupling of the mechanical effectiveness with sEMG triggers revealed the vibration prior to the grip opening (Supplementary Fig. 23d), subsequent grip opening with continuing neuron firing but no obvious frequency shift or amplitude decay (Supplementary Fig. S24b), as well as the muscular congestion that could suggest fatigue history (Supplementary Fig. 23a-c).

Unlike previously adopted cognition implication to avoid the influence of muscle fatigue on active prosthetic control, here we find that nervous fatigue indicated by sEMG seemed to occur ahead of the cognitive perception by the subjects, followed by the metabolic fatigue. Nervous fatigue might not necessarily lead to the change of mechanical output and the gesture change (*e.g.*, grip release), till the occurrence of metabolic fatigue. Therefore, the locally-coupled electromechanical interface can simultaneously indicate the gesture status and muscle fatigue types, providing an advantageous correction measure.

Supplementary Figure 23 | Nervous and metabolic muscle fatigue during consecutive 40 kg “grip till fail” tasks. a-c, Plots show the signatures during the three consecutive grips with 1 min interval in between. d, Plot shows the window of interest in (c), revealing the mechanical vibration initiating the gesture failure and opening of the grip, which was caused by the metabolic muscle fatigue. e, Graph shows the time-frequency analysis of sEMG signals in (d). No obvious shift was shown during the opening stage. Data obtained from subject 5, representative of two sets of consecutive grips.

Supplementary Figure 24 | Nervous fatigue and metabolic muscle fatigue during consecutive 20 kg “grip till fail” tasks. a-b, Plots (*upper row*) show the signatures during the two consecutive grips with 1 min interval in between. Graph (*lower row*) shows the time-frequency analysis of sEMG signals in (d). In the second grip, a relatively high level of sEMG was observed when the subject was resisting the failure of the grip gesture, whereas the drop of the mechanical strain revealed the metabolic muscle fatigue. After the resisting stage, no obvious shift in frequency or drop in the signal amplitude was observed. Data obtained from subject 2, representative of two sets of consecutive grips.

Reviewer #3 (Remarks to the Author):

The authors have even further improved the quality of the manuscript and added interesting and valuable additional data. I recommend acceptance of the manuscript as is.

Response: We appreciate the reviewer for the kind compliments on our revisions. We equally appreciate the reviewer for offering meaningful and constructive suggestions for the improvements.

On minor comment (if the authors want still to include/answer, but it is not necessary): I think they slightly misunderstood my question idea here: "- did the authors do a comparison on adhesion directly on a film with and without these holes? So what do they give in addition to the cracks alone and to a flat gold film? Response: We thank the reviewer for the insightful comments. We have compared the gel adhesion on a flat gold film (Fig. 2g) and found that the adhesion was rather poor (below 10 N/m). This showed that the penetration of gel would be necessary for promoting interlayer adhesion. Which was not aiming at a total smooth film but a film with cracks but no "holes". But maybe this also means that in all cracked films also the holes appear, so it might be hard to separate the influence of these phenomena (though with the new introduced spray technique, the authors could have a look at a "just holes no cracks" film to try to separate the influence.

Response: We thank the reviewer for the kind suggestion. We have also considered to decouple the contribution of microcracks and “holes” with gel penetration and therefore the strong binding. As shown in Supplementary Fig. 2, to exclude the “holes”, we will need to change the elasticity or adhesiveness of the elastomer substrate. However, this will either change the substrate mechanics or the microcrack size. On one hand the change in substrate mechanics will affect the crack propagation therefore the binding strength. On the other hand, the large cracks can also allow the gel penetration, though to a lesser extent (Fig. 2c,d). Therefore, in the peel-off tests, we took the gel/Ti/Au@glass as the negative control to exclude the microcracks and gel penetration. In Fig. 2g, we would like to deliver the message that both the chemical bond between metallic film and the penetration of the gel into the regions of Ti@elastomer layer (correspondingly the contact splitting) would contribute to the strong inter-layer adhesion.

REVIEWERS' COMMENTS:

Reviewer #2 (Remarks to the Author):

I thank the authors for a detailed answer to my comments. From these answers it is evident that the main focus is on the development of the multimodal sensor system, which is valuable. The actual man-machine interfacing aspects are less convincing from the current work (even with the latest revisions) and are left for future work exploiting the proposed system (as the authors argue in the replies). Within this focus, the paper is acceptable as it is.

Reviewer #3 (Remarks to the Author):

I'm fine with the revisions made by the authors and recommend acceptance in the present form.

Detailed Responses to Referees' Comments

We are grateful to the reviewers again for their time and efforts. Their comments are inspiring and the point-to-point responses to the reviewers' comments are detailed below.

Reviewer #2 (Remarks to the Author):

I thank the authors for a detailed answer to my comments. From these answers it is evident that the main focus is on the development of the multimodal sensor system, which is valuable. The actual man-machine interfacing aspects are less convincing from the current work (even with the latest revisions) and are left for future work exploiting the proposed system (as the authors argue in the replies). Within this focus, the paper is acceptable as it is.

Response: We appreciated very much for the positive comments on our revisions. We also agree with the reviewer that it is worthy of digging deep towards improved human-machine interfacing, both with regards to the hardware (sensors) end and the software (algorithms) end.

Reviewer #3 (Remarks to the Author):

Revisions and answers are fine. The revised version becomes acceptable.

Response: We thank the reviewer for the accreditation on this manuscript.